# Pro-Inflammatory Markers in Serum and Saliva in Periodontitis and Hypertension

**DOI:** 10.3390/medicina61061024

**Published:** 2025-05-31

**Authors:** Teodora Bolyarova, Lyubomir Stefanov, Emilia Naseva, Konstantin Stamatov, Samuil Dzhenkov, Blagovest Stoimenov, Ralitsa Pancheva, Nikolay Dochev, Nikolay Ishkitiev

**Affiliations:** 1Department of Periodontology, Faculty of Dental Medicine, Medical University of Sofia, 1431 Sofia, Bulgaria; l.stefanov@fdm.mu-sofia.bg (L.S.); nikido@abv.bg (N.D.); 2Department of Health Management and Health Economics, Faculty of Public Health “Prof. Tzecomir Vodenitcharov, MD, DSc”, Medical University of Sofia, 1524 Sofia, Bulgaria; e.naseva@foz.mu-sofia.bg; 3Medical Faculty, Sofia University St. Kliment Ohridski, 1504 Sofia, Bulgaria; 4Department of Dental, Oral, and Maxillofacial Surgery, Faculty of Dental Medicine, Medical University of Sofia, 1431 Sofia, Bulgaria; k.stamatov@fdm.mu-sofia.bg (K.S.); s.dzhenkov@fdm.mu-sofia.bg (S.D.); 5Clinic of Maxillofacial Surgery, University Hospital Alexandrovska, 1431 Sofia, Bulgaria; 6Department of Propaedeutics of Internal Diseases, Medical Faculty, Medical University of Sofia, 1431 Sofia, Bulgaria; stoimenov90@gmail.com (B.S.); ralitsa.pancheva@abv.bg (R.P.); 7Clinic of Propaedeutics of Internal Diseases, University Hospital Alexandrovska, 1431 Sofia, Bulgaria; 8Department of Medical Chemistry and Biochemistry, Medical Faculty, Medical University of Sofia, 1431 Sofia, Bulgaria; nishkitiev@medfac.mu-sofia.bg

**Keywords:** periodontitis, hypertension, NLRP3, IL1-β

## Abstract

*Background and Objectives*: Over the past few decades, a substantial body of evidence has linked periodontitis to systemic diseases—including hypertension—but the mechanisms underlying this association are not fully understood. This study aims to identify the factors that may mediate this relationship, including an analysis of the inflammatory biomarker NLRP3 and IL-1β levels in serum and saliva in patients with both diseases. *Materials and Methods*: This study included 108 individuals (mean age, 47.8 years, SD 12.8), 38.9% male and 61.1% female. The participants were divided into four groups: Group I—26 healthy participants; Group II—24 participants with periodontitis; Group III—26 participants with hypertension; and Group IV—32 participants with both periodontitis and hypertension. Clinical examinations were performed to diagnose hypertension and periodontitis, including a survey and blood tests in all patients. NLRP3 and IL-1β levels in serum and saliva were measured using ELISA. *Results*: Patients with periodontitis and hypertension were significantly older than those without these conditions (respectively, *p* < 0.001 and *p* < 0.001) and had more missing teeth (respectively, *p* < 0.001 and *p* = 0.037). Higher values were found in the periodontitis and hypertension group than in healthy individuals for VLDL (*p* = 0.001), triglycerides (*p* = 0.001), CRP (*p* = 0.003), WBC (*p* = 0.007), blood sugar (*p* = 0.002), total cholesterol (*p* = 0.003), and LDL (*p* = 0.010). Significantly higher levels of NLRP3 in saliva (*p* = 0.038) and serum (*p* = 0.021) were observed in patients with periodontitis than in those without periodontitis. Significant correlations were found between serum NLRP3 levels and the presence of hypertension (*p* = 0.001) and between saliva IL-1β levels and the presence of hypertension (*p* = 0.010). Serum NLRP3 levels demonstrated a predictive value for hypertension (AUC 0.693, 95% CI 0.590–0.796, and *p* = 0.001), with an established cutoff value of 0.68 ng/mL (sensitivity 0.623, specificity 0.630). *Conclusions*: The higher levels and correlations of pro-inflammatory markers in serum and saliva observed in patients with periodontitis and hypertension support the hypothesis of a relationship between these diseases, likely mediated by low-grade systemic inflammation.

## 1. Introduction

Hypertension is a chronic non-communicable disease with a significant impact on global mortality. The overall prevalence of hypertension among adults has been estimated to range between 30% and 45%. This high prevalence is consistent with global trends across populations, regardless of income levels, affecting countries with low, middle, and high incomes alike [1]. Furthermore, a substantial proportion of individuals receiving anti-hypertensive treatment still fail to achieve therapeutic goals despite their medication. According to the 2018 guidelines by the European Society of Hypertension/European Society of Cardiology, only 38.9% of treated hypertensive patients successfully control their condition and reach target blood pressure values [2].

Despite its high prevalence, the mechanisms underlying the development of arterial hypertension are not completely understood. Recent efforts have increasingly focused on identifying potential risk factors for hypertension, aiming to reduce its prevalence and mitigate its impact. Clinical evidence underscores the significant role of systemic inflammatory responses in the pathogenesis of hypertension [1]. Inflammatory diseases are recognized as significant risk factors for cardiovascular diseases, including hypertension, alongside traditional risk factors [2].

Periodontitis is a chronic multifactorial inflammatory disease associated with a dysbiotic plaque biofilm and characterized by the progressive destruction of the tooth-supporting apparatus. The primary features of periodontitis include the loss of supporting tissues, manifested as clinical attachment loss (CAL) and radiographically confirmed bone loss; the presence of periodontal pockets; and bleeding on probing [3,4]. According to a nationally representative study in the United States, 42.2% of adults aged ≥30 years have some form of periodontitis, including 7.8% with severe periodontitis [5]. Severe periodontitis is the sixth most prevalent disease globally [6]. It poses a significant public health challenge due to its high prevalence, potential to cause tooth loss and disabilities that negatively impact chewing function and aesthetics, and capacity to contribute to social inequalities and deteriorate the quality of life [3].

In recent decades, substantial evidence has linked periodontitis with systemic diseases, aging, reduced quality of life, and premature mortality. Increasing numbers of multidisciplinary consensus reports and collaborative workshops have highlighted the associations between periodontal diseases and various systemic conditions, such as cardiovascular diseases, diabetes mellitus, and respiratory diseases [7]. A published meta-analysis showed that periodontitis (moderate to severe combined diagnosis) is associated with hypertension [8]. Prospective studies have confirmed that a periodontitis diagnosis increases the likelihood of developing hypertension. Patients with periodontitis exhibit higher average systolic and diastolic blood pressure than individuals without periodontitis [8]. Data from observational studies suggest that periodontitis is associated with an increased probability of hypertension. Specific parameters of periodontitis have been identified as closely related to hypertension, including bleeding on probing, periodontal pocket depth, clinical attachment loss, and gingival recession. Furthermore, a group of modifiable risk factors is shared between periodontitis and major non-communicable diseases (cardiovascular diseases, cancer, chronic respiratory diseases, and diabetes). Patients with periodontitis demonstrate not only inflammation of periodontal tissues but also endothelial dysfunction, increased bacterial load (dissemination of endotoxins and exotoxins), metabolic dysregulation, and systemic inflammation. Experimental and observational studies support the important role of systemic inflammation in both the onset and progression of hypertension. Although hypertension is associated with periodontitis, the exact mechanisms mediating this link remain unknown. Considering that systemic inflammatory biomarkers such as C-reactive protein (CRP) and leukocyte count correlate with both periodontitis and hypertension, it is hypothesized that systemic inflammation may mediate the relationship between the two diseases. Confirmation of this link is anticipated from studies focusing on mediators of inflammation [9,10,11].

The pro-inflammatory components NOD-like receptor family pyrin domain-containing 3 (NLRP3) and IL-1β activate inflammation in the periodontal tissues and stimulate the breakdown of periodontal tissue. Additionally, they participate in systemic inflammation, which affects endothelial dysfunction, vascular changes, and the development of arterial hypertension. NLRP3, also known as cryopyrin, is a cytosolic receptor activated by signals mediated by damage-associated molecular patterns (DAMPs) via the inflammasome. Upon activation, NLRP3 triggers the recruitment and autoproteolytic activation of pro-caspase-1, which activates the pro-inflammatory interleukins IL-1β and IL-18. These cytokines can induce low-grade inflammation and subsequent tissue damage. Additionally, they can affect vascular cells, leading to vascular wall remodeling and vasoconstriction, which may contribute to elevated blood pressure [12].

Preliminary data suggest that NLRP3 may be an independent predictor of disease risk in the preclinical stages of certain inflammatory conditions. Evidence indicates that NLRP3 promotes the destruction of periodontal tissues and causes changes in endothelial wall integrity, posing a high risk for the development of endothelial dysfunction [7]. The NLRP3 inflammasome plays a crucial role in hypertension. NLRP3 components are highly expressed in organs that regulate blood pressure, including the kidneys, blood vessels, heart, hypothalamus, and amygdala [13].

Interleukin-1 (IL-1) is a family of pro-inflammatory cytokines with powerful immunoregulatory functions in chronic periodontitis. IL-1α and IL-1β can be released during cellular damage and the activation of immune cells, such as macrophages. In chronic periodontitis, IL-1 and the pro-inflammatory cytokine TNF are believed to regulate the spread of the inflammatory front into deeper areas of connective tissue, leading to attachment loss, osteoclast activation, and subsequent alveolar bone loss [14]. Cytokines from the IL-1 family are considered early mediators of inflammation and potential contributors to the pathogenesis of hypertension. The pro-hypertensive role of IL-1β and IL-18 is associated with their direct effect on the vascular wall. Evidence suggests that IL-1β and IL-18 levels in blood and vascular tissues are elevated in hypertension [15].

These properties of NLRP3 and IL-1β highlight the growing interest in their study to elucidate the link between periodontitis and hypertension, as well as their potential as preclinical biomarkers for the risk of developing both conditions.

Clinical and laboratory investigations would be relevant to confirm the association between periodontitis and hypertension. Pro-inflammatory factors such as NLRP3 and IL-1β, which are linked to local inflammation in periodontitis, may stimulate systemic inflammation and contribute to the development of hypertension. Their quantitative determination in biological fluids—serum and saliva—could confirm the potential influence of periodontitis on the onset of hypertension. Investigating demographic factors and harmful habits may help to identify common risk factors in the pathogenesis of both diseases as part of the mechanisms underlying their association.

The present study aims to assess the existence of an association and the factors that may mediate the relationship between periodontitis and hypertension. It also aims to analyze NLRP3 and IL-1β levels in the saliva and serum of patients with periodontitis and hypertension.

## 2. Materials and Methods

This was a cross-sectional observational study in which clinical data, questionnaire responses, and samples for laboratory analysis were collected simultaneously during a single visit for each participant.

We included 108 individuals aged over 18 years, referred by a general practitioner who visited the cardiology outpatient clinic at “Alexandrovska” University Hospital, Medical University of Sofia, for the diagnosis and treatment of hypertension. They were consecutively enrolled. Of these, 42 were men (38.9%), and 66 were women (61.1%), distributed into four groups: Group I—26 healthy controls; Group II—24 patients with periodontitis; Group III—26 patients with hypertension; Group IV—32 patients with both hypertension and periodontitis. All participants received information about this study and signed an informed consent form. The study was conducted according to the guidelines of the Declaration of Helsinki and approved by the Institutional Ethics Committee of the Medical University of Sofia.

Exclusion criteria for the study were as follows: (1) use of medications such as anti-inflammatory drugs, antibiotics, contraceptives, antioxidants, immunosuppressants, or drugs associated with gingival overgrowth within the last three months before the study; (2) patients with diabetes or prediabetes, obesity (body mass index [BMI] ≥ 30 kg/m^2^), neurological disorders (e.g., multiple sclerosis, Alzheimer’s disease), Crohn’s disease, gastrointestinal disorders, anemia, chronic pulmonary diseases, rheumatoid diseases, autoimmune diseases, chronic liver diseases, and malignancies; (3) pregnancy or breastfeeding; (4) cerebrovascular diseases (including transient ischemic attack or stroke), acute or chronic heart failure, atrial fibrillation, coronary artery disease, acute coronary syndrome, acute myocardial infarction, peripheral vascular disease, severe cardiovascular valve disease, or severe cardiac arrhythmia; (5) secondary hypertension, including chronic renal failure, renal artery stenosis, Conn’s syndrome, or pheochromocytoma; (6) periodontal treatment within the last six months before the study or oral mucosal lesions.

All patients underwent routine clinical evaluations, including medical history, physical examination, and auscultation. Blood pressure was measured three times in the clinical setting or through 24 h ambulatory blood pressure monitoring (ABPM) to diagnose essential hypertension.

Under the following conditions, the diagnosis of hypertension was established according to the 2023 ESH Guidelines, which are consistent with the 2018 ESC/ESH Guidelines, depending on the measured blood pressure (BP) in mmHg: for office BP measurement—systolic BP (SBP) ≥ 140 and/or diastolic BP (DBP) ≥ 90; for 24 h ABPM average—SBP ≥ 130 and/or DBP ≥ 80; for daytime (awake) ABPM average—SBP ≥ 135 and/or DBP ≥ 85; for nighttime (asleep) ABPM average—SBP ≥ 120 and/or DBP ≥ 70 [2,16,17].

Additionally, patients with essential hypertension were categorized into subgroups based on the classification criteria for hypertension: subgroup with high normal BP (SBP 130–139 and/or DBP 85–89); subgroup with stage 1 hypertension (SBP 140–159 and/or DBP 90–99); subgroup with stage 2 hypertension (SBP ≥ 160 and/or DBP ≥ 100) [17].

All patients completed a questionnaire collecting information about their education, socioeconomic status, harmful habits (smoking and excessive alcohol consumption), and the presence of stress. To determine the severity of periodontitis, clinical methods were used to assess periodontal status, including the Hygiene Index (HI); probing pocket depth (PPD); bleeding on probing (BOP); clinical attachment loss (CAL) for each periodontal unit; and the furcation index for teeth with two or three roots. Stages and grades of periodontitis were determined following the New Classification of Periodontal and Peri-Implant Diseases and Conditions, 2017 [3].

Fasting peripheral blood samples were collected from all participants via venipuncture in the morning between 8:00 and 10:00 a.m. Using standard laboratory techniques, venous blood samples were used for routine clinical laboratory testing to determine hematological and biochemical parameters, including complete blood count, serum glucose levels, lipid profile (triglycerides, HDL cholesterol, LDL cholesterol, VLDL cholesterol, and total cholesterol), and C-reactive protein (CRP) levels. Another portion of the blood samples was used for the ELISA determination of NLRP3 and IL-1β concentrations. Blood was collected in vacutainer tubes, kept on ice, and transported immediately to the biochemistry laboratory. Samples were centrifuged at 3000 rpm for 15 min, and the supernatant serum was frozen at –80 °C until analysis.

During the visit in which saliva was collected, no oral interventions were performed prior to sampling. Unstimulated whole saliva was collected according to the method described by Navazesh [18]. Collection took place in the morning (between 9:00 and 11:00 a.m.). The procedure was as follows: participants were asked not to perform oral hygiene procedures (e.g., tooth brushing or mouth rinsing), eat, drink, or chew gum for one hour before saliva collection. All individuals rinsed their mouths with tap water (10 mL) for 30 s and then expelled it before saliva collection. Five minutes after this mouth rinse, each participant was asked to drool saliva into a sterile container at least once per minute for a period of 10 min. The aim was to collect a target volume of approximately 2 mL of whole saliva. Saliva containers were kept on ice during collection and transportation to the laboratory. In the biochemistry laboratory, the saliva samples were centrifuged at 2600× *g* for 15 min at 4 °C. The supernatant was transferred into a new cryovial containing a lyophilized protease inhibitor solution (SigmaFast Protease Inhibitor, Sigma-Aldrich Co., St. Louis, MO, USA). In total, 1 µL of protease inhibitor was added per 1 mL of saliva. All samples were stored at –80 °C until analysis.

Quantitative determination of NLRP3 and IL-1β was performed on both serum and saliva using specific enzyme-linked immunosorbent assay (ELISA) kits: Human Interleukin 1 Beta (IL1β) ELISA Kit and Human NACHT, LRR, and PYD domains-containing protein 3 (NLRP3) ELISA Kit (Abbexa Ltd., Cambridge, UK).

### Statistical Analysis

The results are presented as numbers/proportions for categorical variables. Normally distributed numerical variables are reported as mean and SD; non-Gaussian variables are presented as median and interquartile ranges (IQR: 25th and 75th percentiles). The distribution was assessed using the Kolmogorov–Smirnov test. The Pearson chi-square test (and Fisher’s exact test when applicable) was used to compare categorical variables. The numerical variables of the two groups were compared using the independent samples t-test or Mann–Whitney U-test. More than two groups were compared using one-way ANOVA (Bonferroni for post hoc) or the Kruskal–Wallis test (Bonferroni correction for multiple tests). Spearman’s correlation coefficient (rho) was used to assess the correlation between numerical variables. The diagnostic significance of NLRP3 and IL-1β in serum and saliva was assessed using ROC curves with cutoff value selection at the maximum Youden index. Multivariate logistic regression was used to assess the predictors of binary outcomes. All results were considered significant if *p* < 0.05. The analyses were performed using IBM SPSS v. 26.

## 3. Results

### 3.1. Characteristics of All Four Clinical Groups

The four clinical groups showed similarities regarding gender, education, socioeconomic status, and stress levels (*p* > 0.05) (Table 1). The mean age of the patients was 47.8 years (SD 12.8), with a significant difference between the mean ages across the four groups (*p* < 0.001). Healthy individuals were the youngest (mean 36.3, SD 10.1), followed by patients with hypertension (mean 43.7, SD 11.4), patients with periodontitis (mean 50.0, SD 9.3), and those with both periodontitis and hypertension (mean 58.6, SD 8.3). Significant pairwise differences were observed between the groups, except between the hypertension and periodontitis groups, which were similar in age (Table 2).

Among the patients with hypertension (groups III and IV), 37 individuals (63.8%) had stage 1 hypertension, and 21 individuals (36.2%) had stage 2 hypertension (Table 1).

Among individuals with periodontitis (groups II and IV), 6 (10.71%) had stage I periodontitis, 16 (28.57%) had stage II, 28 (50%) had stage III, and 6 (10.71%) had stage IV periodontitis. In terms of progression risk among patients with periodontitis, 6 individuals (10.71%) were classified as Grade A, 19 (33.93%) as Grade B, and 31 (55.36%) as Grade C (Table 1). An inverse correlation was observed between patient age and the periodontitis grade (Spearman’s rho = −0.321; *p* = 0.007).

Regarding smoking, a significant difference was found among the groups (*p* = 0.021), with the highest proportion of non-smokers among healthy individuals (80.8%), followed by those with hypertension (61.5%), periodontitis and hypertension (59.4%), and periodontitis alone (33.3%) (Table 1). A significant difference in the number of missing teeth between the groups was identified, with the highest number observed in the periodontitis and hypertension group—a median of two teeth (*p* = 0.001). Pairwise comparisons revealed significant differences in the number of missing teeth between healthy individuals and those with periodontitis (*p* = 0.039), as well as between healthy individuals and those with both hypertension and periodontitis (*p* = 0.001).

More missing teeth were observed among individuals with secondary education than among those with higher education (*p* = 0.001), and a moderate correlation between the number of missing teeth and smoking status was found (categorized as non-smokers, former smokers, moderate, or heavy smokers) (Spearman’s rho = 0.506, *p* = 0.05).

Regarding blood parameters, significant differences were observed between the four clinical groups for glucose levels (*p* = 0.001), total cholesterol (*p* = 0.001), and LDL (*p* = 0.001), with the highest levels in the periodontitis and hypertension group (Table 1). Pairwise comparisons showed significantly higher glucose (*p* = 0.002), total cholesterol (*p* = 0.003), and LDL levels (*p* = 0.010) in patients with periodontitis and hypertension than in healthy individuals and significantly higher levels than in the hypertension-only group (*p* = 0.012, *p* = 0.004, and *p* = 0.001, respectively), Table 2.

Significant differences among the four groups were also found for VLDL (*p* = 0.002), triglycerides (*p* = 0.002), CRP (*p* = 0.004), and WBC (*p* = 0.008), again with the highest values observed in the periodontitis and hypertension group (Table 1). Pairwise comparison revealed significantly higher levels for all four parameters in the periodontitis and hypertension group than in healthy individuals (*p* = 0.001, *p* = 0.001, *p* = 0.003, and *p* = 0.007, respectively) (Table 2).

NLRP3 concentrations in saliva (*p* = 0.040) and serum (*p* = 0.002) were significantly different among the four groups, with the highest levels observed in the periodontitis and hypertension group (Table 1). Pairwise comparisons showed significantly higher NLRP3 levels in both saliva and serum in the periodontitis and hypertension group than in the healthy group (*p* = 0.024 and *p* = 0.001, respectively) (Table 2).

No significant differences were found between the four clinical groups in terms of IL-1β levels in saliva or serum (*p* > 0.05) (Table 1). We examined the correlation between the two pro-inflammatory markers, NLRP3 and IL-1β, in both saliva and serum with the clinical and blood parameters across all study participants. The following associations were identified: NLRP3 levels in serum significantly increased with age (Spearman’s rho = 0.465, *p* < 0.001), the number of missing teeth (Spearman’s rho = 0.302, *p* = 0.002), total cholesterol (Spearman’s rho = 0.253, *p* = 0.011), and CRP (Spearman’s rho = 0.218, *p* = 0.031).

IL-1β levels in saliva increased with age (Spearman’s rho = 0.316, *p* = 0.004) and WBC count (Spearman’s rho = 0.289, *p* = 0.009). A positive correlation between NLRP3 and IL-1β in saliva was also observed (Spearman’s rho = 0.248, *p* = 0.039).

### 3.2. Periodontitis vs. Non-Periodontitis Groups

Individuals with and without periodontitis were similar regarding gender, education, socioeconomic status, and the presence of stress (*p* > 0.05) (Table 3), and patients with periodontitis were significantly older than those without periodontitis (*p* < 0.001). Smoking status distribution differed between the periodontitis and non-periodontitis groups, with a higher proportion of non-smokers in the healthy group and a higher proportion of heavy smokers among periodontitis patients (*p* = 0.022).

Significant differences were observed based on the presence or absence of periodontitis for the following parameters: number of missing teeth (*p* < 0.001), total cholesterol (*p* < 0.001), LDL (*p* < 0.001), VLDL (*p* = 0.002), triglycerides (*p* = 0.002), CRP (*p* = 0.007), and WBC (*p* = 0.016). All values were significantly higher in patients with periodontitis (Table 3). The comparison of periodontal parameters with certain blood markers showed a significant, though moderate, correlation between WBC and the periodontitis grade (Spearman’s rho = 0.344, *p* = 0.009). A comparison of pro-inflammatory markers (NLRP3 and IL-1β in serum and saliva) with the presence and parameters of periodontitis showed significant differences in NLRP3 levels in both saliva (*p* = 0.038) and serum (*p* = 0.021), with higher levels in patients with periodontitis.

Among patients with periodontitis, no predictive cutoff value could be established for any of the four pro-inflammatory markers regarding the presence or absence of periodontitis (NLRP3 in saliva and serum; IL-1β in saliva and serum).

### 3.3. Hypertension vs. Non-Hypertension Groups

The presence of a hypertension diagnosis showed no clear correlation with gender (Table 4). Patients with hypertension were significantly older (*p* < 0.001) and had more missing teeth (*p* = 0.037), as well as higher levels of VLDL (*p* = 0.009), triglycerides (*p* = 0.013), CRP (*p* = 0.010), and WBC (*p* = 0.031). Among patients with hypertension, an association was observed between increasing hypertension severity and higher CRP levels (Spearman’s rho = 0.300; *p* = 0.022). We found significant differences in serum NLRP3 levels between patients with and without hypertension, with higher levels in hypertensive individuals (*p* = 0.001). We also detected a significant difference in salivary IL-1β concentrations between patients with and without hypertension, with higher levels in those with the condition (*p* = 0.010).

Of the four pro-inflammatory markers, only serum NLRP3 showed prognostic value for arterial hypertension (AUC 0.693, 95% CI 0.590–0.796, *p* = 0.001), with an established cutoff value of 0.68 ng/mL and above (sensitivity 0.623, specificity 0.630).

We performed multivariate binary logistic regression in the groups (outcome variable: hypertension vs. non-hypertension) with variables identified as significantly different: VLDL, triglycerides, CRP, WBC, NLRP3 (serum), and IL-1β B (saliva). Forward selection revealed NLRP3 (serum) and VLDL as significant. The analysis was then repeated, controlling for age, smoking status, and number of missing teeth. A 1 mmol/L increase in VLDL resulted in a 7-fold increase in the odds for hypertension (OR 7.035, 95% CI 1.223–40.477, *p* = 0.029), while a serum 1 ng/mL increase in NLRP3 increased the odds 1.6-fold (OR 1.616, 95% CI 1.161–2.248, *p* = 0.004).

## 4. Discussion

Large systematic reviews and meta-analyses have shown that periodontitis increases the risk of hypertension and is associated with elevated systolic and diastolic blood pressure. Genetic and clinical studies support a causal role of periodontitis in hypertension, although the mechanisms underlying this association remain unclear [19].

When investigating risk factors for periodontitis, the highest risk tends to be observed among older adults, males, minority racial/ethnic groups, lower-income and less educated groups, and (particularly) smokers [5]. A study conducted on the adult population in Rwanda demonstrated the importance of multiple factors in the pathogenesis of hypertension, including demographic factors, lifestyle, health conditions, and periodontitis. Age emerged as a significant risk factor for hypertension, with risk substantially increasing in older age groups. The gradual rise in blood pressure with age may be due to structural and functional changes in the vascular system, such as arterial sclerosis and endothelial dysfunction [11].

Our results demonstrated that aging is a significant risk factor for both conditions under investigation. Our findings confirmed a higher mean age among patients with periodontitis than among those without, as well as among patients with hypertension compared with non-hypertensive individuals. In patients with periodontitis, we observed an inverse correlation between age and the grade of periodontitis. This fact stems from the significance of this grade as defined by the New Classification of Periodontal and Peri-Implant Diseases and Conditions (2017) [3], which reflects the rate of disease progression over time. According to this classification, younger patients with severe periodontitis tend to exhibit a higher progression rate than older patients with similarly severe disease. Consequently, the grade is higher in younger individuals.

These results are consistent with data from the literature, where authors have reported a higher incidence of hypertension among older individuals, although it is difficult to entirely separate age as a risk factor for vascular changes. Patients aged 30 to 45 years with periodontitis show decreased arterial caliber, suggesting both aging and vascular changes [20].

Our results showed that the clinical groups of patients with periodontitis and hypertension, as well as those with periodontitis alone, had the highest prevalence of smoking. Smoking was more frequent among patients with periodontitis than among individuals without periodontitis. Tobacco use remains a global health issue, affecting 1.3 billion people and causing more than 7 million deaths each year, mainly due to diseases directly related to smoking, including lung cancer, chronic obstructive pulmonary disease, and cardiovascular diseases. Smoking is a major risk factor for the development and progression of periodontal disease. The risk of developing destructive periodontal disease is 5 to 20 times higher in smokers compared with those who have never smoked. Additional evidence for the harmful effects of smoking includes a dose-dependent relationship between the number of cigarettes smoked and the severity of periodontal disease [21]. Furthermore, lifestyle factor studies have reported that individuals with a history of smoking have a 4.10-fold higher risk of developing hypertension than non-smokers [11].

Across all clinical groups, we found a significantly higher number of missing teeth in the group of patients with both periodontitis and hypertension. Among patients with periodontitis, there was a significantly higher number of missing teeth than among individuals without periodontitis. According to the current classification of periodontal diseases, the number of missing teeth can indicate the periodontitis stage, with a higher number of missing teeth associated with greater disease severity. Tooth loss is a primary indicator of periodontitis, as the progression of periodontal disease leads to the destruction of periodontal tissues. We also found that patients with hypertension had significantly more missing teeth than non-hypertensive individuals. Other authors have reported a greater number of missing teeth in individuals with higher blood pressure levels, noting that tooth loss may be caused not only by periodontitis but also by endodontic lesions, fractures, trauma, and caries [20].

We share the view that tooth loss in patients with periodontitis and hypertension cannot be attributed solely to periodontitis, as we did not find a correlation between periodontal parameters and the presence of hypertension. Results from previous studies have strengthened the evidence for an association between periodontitis and blood pressure, based on an increase in periodontal parameters such as probing depth (PD) and clinical attachment loss (CAL), which reflect the severity of periodontitis and influence blood pressure levels [22]. We compared multiple blood markers reflecting systemic inflammation and lipid profiles among patients in the four clinical groups, finding statistically significant differences between them. The highest concentrations of CRP, WBC, total cholesterol, LDL, VLDL, and triglycerides were observed in the group of patients with both hypertension and periodontitis. Patients with periodontitis showed significantly higher concentrations of total cholesterol, LDL, VLDL, triglycerides, CRP, and WBC than those without periodontitis, and WBC values were associated with the grade of periodontitis.

There is strong evidence from cross-sectional studies that plasma CRP levels are elevated in patients with periodontitis compared with control groups, with many studies reporting CRP levels > 2.1 mg/L. There is also moderate evidence regarding the effect of periodontal therapy in reducing CRP levels [23,24]. Scientific evidence indicates that chronic infections are major initiators of chronic inflammation and associated cardiometabolic changes. A previous study confirmed the association of periodontal and peri-implant tissue inflammation with systemic inflammation in patients with hypertension, showing statistically significant high CRP levels in participants with periodontitis, periodontitis combined with mucositis, or peri-implantitis compared with healthy individuals. This association was independent of age, gender, smoking status, and body weight differences [25]. However, biomarkers of inflammation are elevated in individuals with hypertension, including high-sensitivity C-reactive protein (hs-CRP), various cytokines, and complement pathway products [26]. In our study, patients with hypertension exhibited higher levels of VLDL, triglycerides, CRP, and WBC than individuals without hypertension. Recent studies have similarly reported significantly elevated levels of these blood markers (VLDL, triglycerides, CRP, and WBC) in hypertensive patients compared with normotensive individuals, with CRP levels correlating with the severity of hypertension. Key components of the immune and inflammatory pathogenesis of periodontitis significantly overlap with the immune mechanisms underlying hypertension. Clinical studies demonstrate that levels of C-reactive protein and white blood cell counts mediate the association between periodontal disease and high blood pressure. In particular, the activation of Th1, Th17, T regulatory cells, and pro-inflammatory monocytes is essential for both conditions [19]. Similar levels of serum inflammatory markers have been observed among participants with high/uncontrolled blood pressure, suggesting a pro-inflammatory environment that has been described independently for both hypertension and periodontitis. One of the routinely measured markers of systemic inflammation, white blood cells, is known to participate in vascular injury and atherosclerosis and is considered a predictor of hypertension and adverse cardiovascular outcomes. It is believed that lymphocytes and neutrophils mediate these associations, with T cells playing a central role in the development of hypertension and associated vascular abnormalities, partially mediated by the release of reactive oxygen species (ROS), tumor necrosis factor (TNF)-alpha, and interferon (IFN)-gamma [27]. In periodontitis, neutrophils play both protective and destructive roles. As a result of their excessive activation, matrix metalloproteinases and ROS are produced, destroying periodontal tissues [28]. Peripheral neutrophils from patients with periodontitis release increased amounts of IL-1β, IL-8, IL-6, and tumor necrosis factor (TNF)-α upon stimulation with periodontal pathogens [24]. One study also provides evidence that the association between periodontitis parameters reflecting inflammation and high/uncontrolled blood pressure (≥130/80 mmHg) is mediated by inflammatory markers such as CRP, white blood cells, and ferritin. The mean serum levels of these inflammatory markers were significantly higher among participants with high/uncontrolled blood pressure than among those with normal/controlled blood pressure [29].

Data analysis from a large epidemiological study demonstrated an association of periodontal parameters and hypertension with systolic blood pressure (SBP), independent of other common cardiovascular risk factors. Systemic inflammation, assessed through two commonly measured biomarkers (hs-CRP and white blood cell count [WBC]), was independently associated with periodontitis, systolic blood pressure, and diagnosed hypertension and was found to play a mediating role in these associations. Mechanisms have been discussed through which CRP and WBC may partially mediate the link between periodontitis and hypertension, including dysregulation of the renin–angiotensin system, increased oxidative stress, and decreased nitric oxide production, leading to increased endothelial stiffness and dysfunction [9]. A cross-sectional study also reported elevated CRP levels in patients with periodontitis and examined endothelial dysfunction caused by the persistent pro-inflammatory status associated with periodontitis [20].

The results of a meta-analysis showed that periodontitis increases the likelihood of dyslipidemia by 15%. Moreover, mean levels of LDL, total cholesterol, and triglycerides were significantly higher among patients with periodontal diseases, while the mean HDL level was significantly lower compared to individuals without periodontal diseases [30].

Two cross-sectional studies conducted on 54,099 Chinese women and 62,957 Chinese men revealed that total cholesterol, LDL-c, and non-HDL-c were positively associated with the prevalence of hypertension, with hypertensive individuals exhibiting higher values for these markers. Additionally, compared with women without hypertension, hypertensive women had high fasting blood glucose levels and low HDL-c levels. Among men, age, body mass index (BMI), and fasting blood glucose were strongly correlated with hypertension prevalence [31,32].

Summary data from a cross-sectional study involving 117,056 Chinese adults without type 2 diabetes indicated that cholesterol index and non-HDL-C are independent risk factors for the prevalence of hypertension and may serve as effective predictive markers for women. Moreover, ROC curve analysis showed that the cholesterol index and non-HDL-C have a good predictive value for assessing hypertension prevalence in women, with cutoff values of 47.94 mg/dL and 134.34 mg/dL, respectively [33].

In our study, we observed significantly different concentrations of NLRP3 in saliva and serum among the four clinical groups, with the highest levels found in patients with both periodontitis and hypertension. These findings are consistent with the well-established role of the NLRP3 inflammasome in the development of vascular dysfunction. Periodontal inflammation is associated with increased NLRP3 expression and IL-1β and IL-18 activation. These cytokines can act on vascular cells, leading to endothelial dysfunction, vasoconstriction, and elevated blood pressure. Serum NLRP3 levels increase with age, which we identified as a risk factor for both periodontitis and hypertension. Furthermore, the correlation of serum NLRP3 levels with the number of missing teeth among all patients highlights the influence of oral health status on this systemic factor. These correlations between serum NLRP3 and inflammation-related factors (total cholesterol and CRP) can be explained by NLRP3’s role in the initiation and progression of the inflammatory process. We also observed higher concentrations of the pro-inflammatory marker IL-1β in saliva with increasing age, likely reflecting the greater prevalence of periodontitis and hypertension in older patients and the more frequent presence of localized inflammation. The positive correlation between salivary NLRP3 and IL-1β suggests a mutual relationship between these two markers—possibly reflecting their increased presence in periodontal tissues due to the ongoing inflammatory process in periodontitis—and their subsequent detection in saliva. Notably, the NLRP3 marker in saliva and serum shows significant differences between the clinical groups studied, whereas IL-1β in saliva and serum does not. A possible explanation for this result is the wide range (IQR) observed for IL-1β in both saliva and serum; this suggests that, given the sample size, it is not possible to establish statistically significant differences between the groups. In contrast, the range (IQR) for the NLRP3 marker in saliva and serum is relatively small, making it possible to detect significant differences between the groups within the studied sample size.

An important component of periodontitis pathogenesis is the increased number of white blood cells. Studies have shown that in moderate and severe periodontitis, there is a significantly higher count of neutrophils, lymphocytes, and total leukocytes [34], while non-surgical periodontal therapy improves clinical parameters and reduces WBC count [35]. The observed correlation between salivary IL-1β levels and WBC counts in our study supports the role of local periodontal inflammation as a factor of systemic significance, leading to changes in the number of immune defense cells.

We found significantly higher concentrations of NLRP3 in both saliva and serum among patients with periodontitis than in those without the disease. We associate these results with the development of a local inflammatory process in periodontal tissues, where the NLRP3 factor increases locally, subsequently appearing in saliva and likely correlating with the severity of the process. As evidence of the systemic impact of periodontitis, we can discuss the correlation between serum NLRP3 levels and the presence of periodontitis. Local increases in NLRP3 during periodontitis likely contribute to its entry into the systemic circulation, promoting systemic inflammation.

Previous experimental studies have demonstrated the NLRP3 inflammasome’s involvement in the development of hypertension and hypertension-induced organ damage, whereas it has been associated with blood pressure reduction and renal protection in hypertension [12]. Periodontitis activates the NLRP3 inflammasome in serum and saliva under conditions of comorbidity, with studies indicating that patients with periodontitis and periodontitis + type II diabetes have higher concentrations of NLRP3 in serum and saliva than healthy controls and patients with type II diabetes alone [36].

The significance of the pro-inflammatory marker NLRP3 in the pathogenesis of hypertension can also be discussed based on our findings, showing higher serum levels in hypertensive patients than in normotensive individuals. We did not find other clinical studies identifying such a relationship. The determination of a threshold value of 0.68 ng/mL for serum NLRP3 has prognostic relevance for the manifestation of arterial hypertension. Among the pro-inflammatory markers studied, only serum NLRP3 showed prognostic value for arterial hypertension, with an area under the curve (AUC) of 0.693, indicating moderate accuracy in distinguishing hypertensive patients from normotensive individuals. The established cutoff value of 0.68 ng/mL demonstrated a relatively balanced sensitivity (62.3%) and specificity (63.0%), suggesting the potential of serum NLRP3 as a biomarker for identifying patients at risk of hypertension. Its use in combination with other clinical and laboratory parameters may be useful in the diagnosis and risk assessment of hypertension. The use of serum NLRP3 to predict hypertension is based on the central role of inflammation in the pathogenesis of the disease. Our results suggest the need for future studies to establish cutoff values for other pro-inflammatory markers as well, thus allowing for the use of a panel of markers to detect hypertension risk with higher sensitivity and specificity.

In our multivariate binary logistic regression analysis, among all included variables, VLDL and serum NLRP3 emerged as statistically significant predictors of hypertension. When the analysis was repeated while controlling for confounding variables (age, smoking status, and number of missing teeth) the results confirmed the significance of these two markers. Our findings show that an increase in VLDL by 1 mmol/L is associated with a 7-fold increase in the odds of hypertension (controlling for confounding variables), confirming that elevated VLDL levels are strongly linked to hypertension and may serve as an important predictor. The finding that a 1 ng/mL increase in serum NLRP3 raises the odds of hypertension 1.6 times is statistically significant (controlling for confounding variables) and supports the hypothesis that the pro-inflammatory marker NLRP3 plays a role in the pathogenesis of hypertension. These results may be used for the early identification of individuals at increased risk of developing hypertension and for the development of preventive strategies, which is clinically relevant.

Interleukin-1 (IL-1) plays a leading role in mediating the destruction of periodontal tissues, and evidence from previous studies supports its important role in patients with periodontitis: IL-1 levels in gingival crevicular fluid are significantly higher in patients with chronic periodontitis than in periodontally healthy individuals, and particularly high levels have been observed in older patients compared with younger ones. Interventional studies have shown that in patients with periodontitis, non-surgical periodontal treatment significantly reduces IL-1β levels, suggesting that successful therapy may prevent disease progression at least partially by suppressing IL-1β activity [14].

In our study, the pro-inflammatory marker IL-1β in saliva was found at higher concentrations in relation to the diagnosis of hypertension. Our findings are consistent with previous studies reporting increased levels of NLRP3, caspase-1, and IL-1β in the T cells of hypertensive patients [12]. IL-1β and IL-18 concentrations are elevated in the circulation of patients with essential hypertension, and this increase is associated with elevated blood pressure [12]. Several studies have found that patients with essential hypertension have high serum IL-1β levels, suggesting the pro-hypertensive effects of IL-1β. Evidence indicates that IL-1β contributes to the development of hypertension through its inflammation-mediating signaling and by regulating the function of vascular smooth muscle cells and remodeling the extracellular matrix. Research results suggest that IL-1β participates in the pathogenesis of various hypertension types, indicating that similar regulatory mechanisms may be involved in vascular dysfunction despite the different etiologies of the disease [37].

No previous studies have reported an association between salivary IL-1β and the diagnosis of hypertension. The elevated salivary IL-1β concentrations we observed in hypertensive patients could indicate that locally synthesized IL-1β in periodontal tissues can enter the bloodstream, contribute to the systemic pro-inflammatory status, and play a role in the pathogenesis of hypertension.

The positive correlation we found between salivary NLRP3 and IL-1β provides logical confirmation of the known biochemical pathway of NLRP3 activation and the subsequent activation of pro-inflammatory IL-1β. These results strongly support the interconnection between the two macromolecules in the ongoing local inflammatory process and highlight their potential as markers reflecting this process.

### Strengths and Limitations

One limitation of this study is the relatively small sample size of 108 individuals, which did not allow for extensive subgroup analyses. Another limitation is the presence of significant differences in certain demographic characteristics (such as age and smoking status) among the clinical groups. However, it is important to note that the participants were not selectively recruited but were enrolled consecutively. The design of our study is cross-sectional, limiting our ability to draw conclusions regarding the causal mechanisms of the association between periodontitis and hypertension; however, it does identify important associations that can guide future longitudinal studies. Longitudinal studies or studies analyzing the effects of periodontal treatment on potential markers for hypertension are necessary to establish causal relationships between periodontitis and hypertension.

One of this study’s strengths is that it is the first investigation of the pro-inflammatory markers NLRP3 and IL-1β in both serum and saliva in relation to periodontitis and hypertension, as well as its identification of associations supporting the hypothesis of low-grade inflammation’s role in the development of hypertension.

## 5. Conclusions

The widespread prevalence of hypertension at national and global levels presents a major public health issue. Finding effective methods for detecting the risk and presence of the disease would contribute to its early diagnosis.

Our study demonstrated a significantly higher prevalence of risk factors, elevated concentrations of pro-inflammatory markers, and biochemical parameters in both blood and saliva among patients with periodontitis and hypertension. Furthermore, patients with hypertension exhibited significantly more missing teeth than individuals without hypertension. These findings support the determination of the pro-inflammatory marker NLRP3 in serum as a tool to identify individuals at risk of hypertension and to guide actions to manage the condition.

Our results can be used to plan larger-scale investigations in both the Bulgarian population and other populations. They support the significance of periodontitis in the development of hypertension and point toward the need for research aimed at planning health strategies for the evaluation and management of periodontal diseases as part of cardiovascular health control.

## Figures and Tables

**Table 1 medicina-61-01024-t001:** Overall comparison of demographic, clinical, and laboratory parameters across all four clinical groups.

Variables		Healthy Control		Periodontitis		Periodontitis and Hypertension		Hypertension		*p*
		n/Mean/Median	%/SD/IQR	n/Mean/Median	%/SD/IQR	n/Mean/Median	%/SD/IQR	n/Mean/Median	%/SD/IQR	
Sex	Male	11	42.3%	5	20.8%	15	46.9%	11	42.3%	0.221
	Female	15	57.7%	19	79.2%	17	53.1%	15	57.7%	
Age	Mean; SD	36.3	10.1	50	9.3	58.6	8.3	43.7	11.4	<0.001
Education	High school or lower	4	15.4%	10	41.7%	13	40.6%	9	34.6%	0.151
	University or college	22	84.6%	14	58.3%	19	59.4%	17	65.4%	
Socioeconomic status	Low	1	3.8%	1	4.2%	5	15.6%	2	7.7%	0.475
	Middle	23	88.5%	23	95.8%	26	81.3%	23	88.5%	
	High	2	7.7%	0	0.0%	1	3.1%	1	3.8%	
Smoking	Non-smoker	16	61.5%	6	25.0%	9	28.1%	11	42.3%	0.021
	Ex-smoker	5	19.2%	2	8.3%	10	31.3%	5	19.2%	
	Current smoker, 1–20 cigarettes/day	4	15.4%	10	41.7%	5	15.6%	7	26.9%	
	Current smoker > 20 cigarettes/day	1	3.8%	6	25.0%	8	25.0%	3	11.5%	
Smoking	Non-smoker	21	80.8%	8	33.3%	19	59.4%	16	61.5%	0.008
	Current smoker	5	19.2%	16	66.7%	13	40.6%	10	38.5%	
Stress	No	18	69.2%	16	66.7%	24	75.0%	21	80.8%	0.673
	Yes	8	30.8%	8	33.3%	8	25.0%	5	19.2%	
Blood pressure category	Stage 1 hypertension					15	46.90%	22	84.60%	0.003
	Stage 2 hypertension					17	53.10%	4	15.40%	
Periodontitis stage	1			2	8.30%	4	12.50%			0.156
	2			6	25.00%	10	31.30%			
	3			11	45.80%	17	53.10%			
	4			5	20.80%	1	3.10%			
Periodontitis grade	A			1	4.20%	5	15.60%			0.103
	B			7	29.20%	12	37.50%			
	C			16	66.70%	15	46.90%			
CAL max mm	Median; IQR			6	4–8	5	3–7			0.126
N missing teeth	Median; IQR	0	0–0	1	0–9	2	0–6	0	0–3	0.001
PD max mm	PD ≤ 5 mm			11	45.80%	19	59.40%			0.160
	PD 6–7 mm			6	25.00%	10	31.30%			
	PD > 7 mm			7	29.20%	3	9.40%			
PD max mm	Median; IQR			6	4–9	5	5–7			0.574
BoP %	Median; IQR			60	15–100	25	10–60			0.089
Blood sugar mmol/L	Median; IQR	4.61	4.27–4.9	4.63	4.37–4.85	4.99	4.82–5.44	4.82	4.55–5.34	0.001
Total cholesterol mmol/L	Median; IQR	4.73	4.10–5.79	5.24	4.78–5.9	5.65	5.28–6.79	4.83	4.29–5.33	0.001
HDL mmol/L	Median; IQR	1.61	1.16–2.09	1.54	1.27–1.7	1.47	1.08–1.79	1.63	1.25–1.8	0.687
LDL mmol/L	Mean; SD	2.94	0.80	3.26	0.83	3.75	1.19	2.80	0.78	0.001
VLDL mmol/L	Median; IQR	0.30	0.20–0.50	0.55	0.30–0.70	0.55	0.45–0.90	0.40	0.30–0.60	0.002
Triglycerides mmol/L	Median; IQR	0.72	0.48–1.09	1.21	0.66–1.51	1.27	0.92–1.95	0.88	0.66–1.27	0.002
CRP mg/L	Median; IQR	0.8	0.4–1.8	0.9	0.5–2.6	2.6	1.1–4.7	1.2	0.5–2.5	0.004
WBC ×10^9^/L	Mean; SD	5.9	1.4	7.2	1.8	7.5	1.9	7.1	1.9	0.008
NLRP3 ng/mL (saliva)	Median; IQR	6.2	3.5–12.4	14.7	6.7–31.6	8.1	7.2–14.2	8.2	5.0–19.9	0.040
NLRP3 ng/mL (serum)	Median; IQR	0.4	0.1–1.2	0.5	0.2–1.6	2.4	0.5–5.1	0.8	0.3–1.5	0.002
IL-1β B pg/mL (saliva)	Median; IQR	76.5	31.4–270.1	135.1	78.3–291.5	339.7	125.0–460.5	203.8	83.0–263.1	0.055
IL-1β B pg/mL (serum)	Median; IQR	14.2	10.7–18.4	12.1	9.5–18.1	14.3	10.3–15.2	15.0	10.7–16.0	0.822

**Table 2 medicina-61-01024-t002:** Pairwise comparisons of demographic, clinical, and laboratory parameters between the four clinical groups.

*p*-Values for Post Hoc	Age	N Missing Teeth	Blood Sugar mmol/L	Total Cholesterol mmol/L	LDL mmol/L	VLDL mmol/L	Triglycerides mmol/L	CRP mg/L	WBC ×10^9^/L	NLRP3 ng/mL (Saliva)	NLRP3 ng/mL (Serum)
Healthy control—hypertension	0.045	0.637	1.000	1.000	1.000	0.593	0.708	1.000	0.089	1.000	1.000
Healthy control—periodontitis	0.000	0.039	0.534	0.752	1.000	0.306	0.300	1.000	0.057	0.928	0.515
Healthy control—periodontitis and hypertension	0.000	0.001	0.002	0.003	0.010	0.001	0.001	0.003	0.007	0.024	0.001
Hypertension—periodontitis	0.146	1.000	1.000	0.853	0.595	1.000	1.000	1.000	1.000	1.000	1.000
Hypertension—periodontitis and hypertension	0.000	0.157	0.012	0.004	0.001	0.193	0.203	0.117	1.000	0.651	0.063
Periodontitis—periodontitis and hypertension	0.009	1.000	0.473	0.544	0.372	0.589	0.744	0.166	1.000	0.851	0.427

**Table 3 medicina-61-01024-t003:** Comparison of demographic, clinical, and laboratory parameters between the periodontitis and non-periodontitis groups.

Variables		Non-Periodontitis		Periodontitis		*p*
		n/Mean/Median	%/SD/IQR	n/Mean/Median	%/SD/IQR	
Sex	Male	22	42.3%	20	35.7%	0.555
	Female	30	57.7%	36	64.3%	
Age	Mean; SD	40	11.3	54.9	9.7	<0.001
Education	High school or lower	13	25.0%	23	41.1%	0.077
	University or college	39	75.0%	33	58.9%	
Socioeconomic status	Low	3	5.8%	6	10.7%	0.377
	Middle	46	88.5%	49	87.5%	
	High	3	5.8%	1	1.8%	
Smoking	Non-smoker	27	51.9%	15	26.8%	0.022
	Ex-smoker	10	19.2%	12	21.4%	
	Current smoker, 1–20 cigarettes/day	11	21.2%	15	26.8%	
	Current smoker, >20 cigarettes/day	4	7.7%	14	25.0%	
Smoking	Non-smoker	37	71.2%	27	48.2%	0.015
	Current smoker	15	28.8%	29	51.8%	
Stress	No	39	75.0%	40	71.4%	0.676
	Yes	13	25.0%	16	28.6%	
N missing teeth	Median; IQR	0	0–2	1	0–7	<0.001
Blood sugar mmol/L	Median; IQR	4.70	4.36–5.06	4.87	4.62–5.35	0.071
Total cholesterol mmol/L	Median; IQR	4.79	4.19–5.43	5.49	5.00–6.22	<0.001
HDL mmol/L	Median; IQR	1.63	1.22–1.88	1.49	1.27–1.75	0.291
LDL mmol/L	Mean; SD	2.87	0.79	3.55	1.08	<0.001
VLDL mmol/L	Median; IQR	0.4	0.3–0.5	0.6	0.4–0.8	0.002
Triglycerides mmol/L	Median; IQR	0.82	0.62–1.17	1.21	0.84–1.71	0.002
CRP mg/L	Median; IQR	1.0	0.4–2.2	1.8	0.8–4.0	0.007
WBC ×10^9^/L	Mean; SD	6.5	1.8	7.4	1.8	0.016
NLRP3 ng/mL (saliva)	Median; IQR	7.2	4.1–14.3	8.9	6.7–20.6	0.038
NLRP3 ng/mL (serum)	Median; IQR	0.5	0.2–1.3	1.0	0.4–3.8	0.021
IL-1β B pg/mL (saliva)	Median; IQR	123.9	36.4–268.1	164.4	95.2–411.9	0.094
IL-1β B pg/mL (serum)	Median; IQR	14.7	10.7–17.8	13.8	9.8–16.7	0.358

**Table 4 medicina-61-01024-t004:** Comparison of demographic, clinical, and laboratory parameters between the hypertension and non-hypertension groups.

Variables		Non-Hypertension		Hypertension		*p*
		n/Mean/Median	%/SD/IQR	n/Mean/Median	%/SD/IQR	
Sex	Male	16	32.0%	26	44.8%	0.173
	Female	34	68.0%	32	55.2%	
Age	Mean; SD	42.9	11.8	51.9	12.3	<0.001
Education	High school or lower	14	28.0%	22	37.9%	0.275
	University or college	36	72.0%	36	62.1%	
Socioeconomic status	Low	2	4.0%	7	12.1%	0.318
	Middle	46	92.0%	49	84.5%	
	High	2	4.0%	2	3.4%	
Smoking	Non-smoker	22	44.0%	20	34.5%	0.324
	Ex-smoker	7	14.0%	15	25.9%	
	Current smoker, 1–20 cigarettes/day	14	28.0%	12	20.7%	
	Current smoker, >20 cigarettes/day	7	14.0%	11	19.0%	
Smoking	Non-smoker	29	58.0%	35	60.3%	0.895
	Current smoker	21	42.0%	23	39.7%	
Stress	No	34	68.0%	45	77.6%	0.262
	Yes	16	32.0%	13	22.4%	
N missing teeth	Median; IQR	0	0–2	1	0–4	0.037
Blood sugar mmol/L	Median; IQR	4.61	4.31–4.90	4.96	4.68–5.41	0.574
Total cholesterol mmol/L	Median; IQR	5.15	4.30–5.80	5.37	4.60–5.88	0.137
HDL mmol/L	Median; IQR	1.56	1.22–1.95	1.52	1.25–1.80	0.703
LDL mmol/L	Mean; SD	3.09	0.82	3.33	1.13	0.226
VLDL mmol/L	Median; IQR	0.4	0.3–0.6	0.5	0.3–0.8	0.009
Triglycerides mmol/L	Median; IQR	0.93	0.56–1.34	1.10	0.75–1.87	0.013
CRP mg/L	Median; IQR	0.9	0.4–2.1	1.7	0.8–3.6	0.010
WBC ×10^9^/L	Mean; SD	6.6	1.7	7.3	1.9	0.031
NLRP3 ng/mL (saliva)	Median; IQR	8.7	4.3–14.8	8.1	5.8–16.5	0.885
NLRP3 ng/mL (serum)	Median; IQR	0.4	0.1–1.5	1.0	0.5–4.2	0.001
IL-1β B pg/mL (saliva)	Median; IQR	107.4	46.5–270.1	248.2	94.3–447.0	0.022
IL-1β B pg/mL (serum)	Median; IQR	13.4	9.7–18.1	14.5	10.6–15.5	0.991

## Data Availability

The data presented in this study are available upon request from the corresponding author.

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
