# Peer review of "Pro-Inflammatory Markers in Serum and Saliva in Periodontitis and Hypertension"

_medicina, 2025, doi:10.3390/medicina61061024_

Round 1
Reviewer 1 Report
Comments and Suggestions for Authors
This is a cross-sectional study aiming to comparatively evaluate the demographic, clinical periodontal parameters, and serum, salivary levels of NLRP3 and IL-1b in study groups with or without periodontitis or hypertension. The study design is scientifically valid and the manuscript is well written. There are some minor issues to be addressed by the authors before the manuscript can be accepted for publication.
- Please spell out all abbreviations in the abstract
- Page 2, line 68: please delete “with existing teeth”
- The introduction part needs to be shortened particularly the two paragraphs about ILs can be condensed.
- It will be nice to provide a hypothesis just before stating the aim of the study.
- Please correct the typo on page 4, line 169: Crohn’s Disease
- Please clarify the relevant harmful habits on page 4, line 186.
- Please shorten the discussion by eliminating those sentences that are not directly related with the study content such as those related with DM.
- The major limitation is that the study groups were not balanced with regard to the age and smoking status and this needs to be clearly stated at the end of the discussion.
Author Response
Dear Reviewer,
Thank you for your careful reading and helpful comments.
We have made the necessary revisions in accordance with your recommendations, as outlined below:
Comments 1: Please spell out all abbreviations in the abstract
Response 1: We have not spelled out all abbreviations in the abstract because they are widely known and well understood by specialists in the field, who constitute the primary target audience of the study. Furthermore, fully spelling out all abbreviations would significantly increase the length of the text and lead to exceeding the required word limit for the abstract. To comply with the word count restrictions, it would be necessary to shorten the content, which would limit the amount and quality of the information provided.
Comments 2: Page 2, line 68: please delete “with existing teeth”
Response 2: It was deleted.
Comments 3: The introduction part needs to be shortened particularly the two paragraphs about ILs can be condensed.
Response 3: The part of the introduction referring to interleukins has been shortened — from line 103 to line 128.
Comments 4: It will be nice to provide a hypothesis just before stating the aim of the study.
Response 4: A hypothesis is formulated before stating the aim of the study — from line 129 to line 139.
Comments 5: Please correct the typo on page 4, line 169: Crohn’s Disease
Response 5: We believe it is written correctly – Crohn’s disease. On line 161.
Comments 6: Please clarify the relevant harmful habits on page 4, line 186.
Response 6: It has been clarified – harmful habits (smoking and excessive alcohol consumption). On line 186.
Comments 7: Please shorten the discussion by eliminating those sentences that are not directly related with the study content such as those related with DM.
Response 7: The discussion has been shortened in the specified section. The focus is on the impact of periodontitis on NLRP3. From line 536 to line 540.
Comments 8: The major limitation is that the study groups were not balanced with regard to the age and smoking status and this needs to be clearly stated at the end of the discussion.
Response 8: This limitation is mentioned in the discussion — from line 605 to line 608.
Reviewer 2 Report
Comments and Suggestions for Authors
This manuscript investigates a clinically important association between periodontitis and hypertension, focusing on the potential mediating role of inflammatory biomarkers, specifically NLRP3 and IL-1β. The study design involves collecting data on these biomarkers, periodontal status, and blood pressure from different patient groups. The methodology for data and sample collection appears generally sound, and the laboratory analyses are standard techniques. The results section presents findings related to group characteristics and biomarker levels. However, several key issues need to be addressed. My detailed comments and recommendations are provided below.
Major Comments:
- The study appears to be cross-sectional, based on the description of data collection at a single time point. While the Introduction correctly mentions that prospective studies have indicated a causal link between periodontitis and hypertension, this specific study design does not allow for inferring causality. The language used throughout the manuscript, particularly in the Introduction and Discussion, should be carefully revised to avoid any implication of causality based solely on these cross-sectional findings. (Recommendation) Explicitly state the cross-sectional nature of the study design in the "Materials and Methods" section. Adjust the language in the Introduction and Discussion to accurately reflect that the study identifies associations or correlations, not causal relationships. Clearly discuss the limitations of a cross-sectional design regarding causal inference.
Results-
In "Results" section indicates significant differences between the study groups regarding crucial factors such as age, smoking status, lipid profiles (total cholesterol, LDL, VLDL, triglycerides), CRP, and WBC count. These are well-established risk factors for both periodontitis and hypertension and act as significant confounders. While the paper presents comparisons between groups, the analysis does not appear to have statistically adjusted for these key variables. This makes it difficult to determine if the observed associations between the diseases and the biomarkers are independent of these confounders. Conduct and present results from multivariable regression analyses (e.g., ANCOVA for continuous outcomes, logistic regression for binary outcomes) to adjust for significant confounders such as age, smoking status, and relevant metabolic/inflammatory markers (lipids, CRP, WBC) when evaluating the association between periodontitis/hypertension and the levels of NLRP3 and IL-1β.
When describing the staging of hypertension, the text states: "Among the patients with hypertension (groups III and IV), 37 individuals (59.5%) had stage 1 hypertension, and 21 individuals (19%) had stage 2 hypertension (Table 1)." The raw numbers (37 and 21) sum to 58, which is the correct total number of hypertensive patients (26 in Group III + 32 in Group IV). However, the percentages provided (59.5% and 19%) do not sum to 100% and do not accurately reflect the proportions of 37 and 21 out of 58 (37/58 63.8%; 21/58 36.2%). This suggests an error in reporting percentages. Verify the calculation of these percentages and correct the values reported in the text and Table 1 (once provided).
The findings regarding the two biomarkers differ. NLRP3 levels (in both saliva and serum) show significant differences among the four clinical groups and are higher in the combined periodontitis+hypertension group compared to healthy controls. Serum NLRP3 is also higher in hypertensive vs. non-hypertensive individuals. However, IL-1β levels are reported as not significantly different among the four main groups (Table 1), although salivary IL-1β is found to be significantly different when comparing the broader categories of hypertensive vs. non-hypertensive individuals (Table 4). This discrepancy between the analyses for IL-1β needs clarification. In the "Discussion," delve deeper into potential reasons for the differences in findings between NLRP3 and IL-1β. Discuss why IL-1β might not have shown significant differences across the four specific groups, considering factors like sample size per group, marker sensitivity, or specific roles of these cytokines in the pathogenesis of these conditions. Ensure consistency in how IL-1β findings are described based on the specific comparisons performed.
The reported significant inverse correlation (Spearman's rho = , ) between patient age and the "periodontitis progression risk grade (Grade)" is counter-intuitive. Generally, the severity and stage of periodontitis tend to increase with age due to cumulative disease progression. Clarify precisely whether this correlation refers to the "grade" (risk of progression, A/B/C) or the "stage" (severity, I/II/III/IV). If the finding of an inverse correlation with age for the "Grade" is accurate, provide a detailed biological or clinical explanation for this observation in the "Discussion," considering how the grading system is applied and potential selection biases in the study population (e.g., perhaps older individuals with less aggressive forms were enrolled).
The "Methods" section mentions assessing the "Diagnostic significance of NLRP3 and IL-1β in serum and saliva" using ROC curves. However, the results of this analysis (e.g., Area Under the Curve - AUC, sensitivity, specificity, optimal cutoff values) are not presented in the "Results" section. This analysis is important for evaluating the potential clinical utility of these biomarkers for identifying periodontitis and/or hypertension. Include the results of the ROC curve analysis for NLRP3 and IL-1β (in both serum and saliva), presenting AUC values, confidence intervals, p-values, and potentially suggested cutoff values, in the "Results" section. Discuss the implications of these findings in the "Discussion."
Minor Comments:
(Introduction )
The Introduction provides a good background. The transition to discussing NLRP3 and IL-1β could be smoother by briefly summarizing their known roles in periodontitis and hypertension individually before proposing their potential role as mediators linking the two conditions.
(M&M)
- Specify how participants were recruited from the cardiology clinic (e.g., consecutively enrolled patients meeting inclusion criteria?).
- Clarify if AHA/ACC 2023 and 2018 guidelines were applied interchangeably or if there's a specific rationale for mentioning both years.
- Refine the description of Saliva Collection. Define if the 2 mL target volume was the primary goal, or if the collection strictly adhered to the 10-minute timeframe, regardless of the final volume. "Drool saliva... at least once per minute" could be clarified regarding the intended flow rate (e.g., unstimulated saliva collection usually aims for a specific resting flow rate). Confirm whether the saliva volume was precisely measured for adding the protease inhibitor (1 µL per mL) or if a fixed amount was added to an estimated volume.
(Discussion)
- Expand the discussion on how the observed elevated NLRP3 levels in the combined disease group specifically align with the known roles of NLRP3 in inflammation and vascular dysfunction related to hypertension.
- Reiterate and emphasize the main limitations of the cross-sectional design clearly in the Discussion.
- While the authors acknowledge other causes of tooth loss (caries, trauma, etc.), briefly discuss if any attempt was made to account for these factors in the data collection or analysis, or acknowledge this as an inherent limitation when using missing teeth as a proxy for periodontitis severity.
- Conclude the Discussion by suggesting specific directions for future research based on the findings, such as longitudinal studies, intervention trials (e.g., assessing the impact of successful periodontal therapy on blood pressure and these biomarkers), or larger studies capable of more robust multivariable modeling
- The manuscript is generally well-written. However, a final proofread by a native English speaker or a professional editing service would help correct minor grammatical issues and awkward phrasing, enhancing clarity and readability.
Author Response
Dear Reviewer,
Thank you for your careful reading and helpful comments.
We have made the necessary revisions in accordance with your recommendations, as outlined below:
Major Comments:
Comments 1: The study appears to be cross-sectional, based on the description of data collection at a single time point. While the Introduction correctly mentions that prospective studies have indicated a causal link between periodontitis and hypertension, this specific study design does not allow for inferring causality. The language used throughout the manuscript, particularly in the Introduction and Discussion, should be carefully revised to avoid any implication of causality based solely on these cross-sectional findings. (Recommendation) Explicitly state the cross-sectional nature of the study design in the "Materials and Methods" section. Adjust the language in the Introduction and Discussion to accurately reflect that the study identifies associations or correlations, not causal relationships. Clearly discuss the limitations of a cross-sectional design regarding causal inference.
Response 1: In the Materials and Methods section, we have indicated the cross-sectional nature of the study — from line 145 to line 147. The discussion reflects that the study demonstrates associations rather than causal relationships. The limitations of the cross-sectional design are indicated — from line 608 to line 611.
Results-
Comments 2: In "Results" section indicates significant differences between the study groups regarding crucial factors such as age, smoking status, lipid profiles (total cholesterol, LDL, VLDL, triglycerides), CRP, and WBC count. These are well-established risk factors for both periodontitis and hypertension and act as significant confounders. While the paper presents comparisons between groups, the analysis does not appear to have statistically adjusted for these key variables. This makes it difficult to determine if the observed associations between the diseases and the biomarkers are independent of these confounders. Conduct and present results from multivariable regression analyses (e.g., ANCOVA for continuous outcomes, logistic regression for binary outcomes) to adjust for significant confounders such as age, smoking status, and relevant metabolic/inflammatory markers (lipids, CRP, WBC) when evaluating the association between periodontitis/hypertension and the levels of NLRP3 and IL-1β.
Response 2: Multivariate logistic regression was used to assess the predictors of the binary outcome and was described in the Methods section - lines 234 to 235. The results are presented at the end of the Results section — from line 337 to line 345. The results are discussed in the Discussion section — from line 558 to line 570.
Comments 3: When describing the staging of hypertension, the text states: "Among the patients with hypertension (groups III and IV), 37 individuals (59.5%) had stage 1 hypertension, and 21 individuals (19%) had stage 2 hypertension (Table 1)." The raw numbers (37 and 21) sum to 58, which is the correct total number of hypertensive patients (26 in Group III + 32 in Group IV). However, the percentages provided (59.5% and 19%) do not sum to 100% and do not accurately reflect the proportions of 37 and 21 out of 58 (37/58 ≈ 63.8%; 21/58 ≈ 36.2%). This suggests an error in reporting percentages. Verify the calculation of these percentages and correct the values reported in the text and Table 1 (once provided).
Response 3: We checked the calculations and corrected the percentages in the text — on lines 251 and 252. We believe that the percentages in the table are accurate.
Comments 4: The findings regarding the two biomarkers differ. NLRP3 levels (in both saliva and serum) show significant differences among the four clinical groups and are higher in the combined periodontitis+hypertension group compared to healthy controls. Serum NLRP3 is also higher in hypertensive vs. non-hypertensive individuals. However, IL-1β levels are reported as not significantly different among the four main groups (Table 1), although salivary IL-1β is found to be significantly different when comparing the broader categories of hypertensive vs. non-hypertensive individuals (Table 4). This discrepancy between the analyses for IL-1β needs clarification. In the "Discussion," delve deeper into potential reasons for the differences in findings between NLRP3 and IL-1β. Discuss why IL-1β might not have shown significant differences across the four specific groups, considering factors like sample size per group, marker sensitivity, or specific roles of these cytokines in the pathogenesis of these conditions. Ensure consistency in how IL-1β findings are described based on the specific comparisons performed.
Response 4: In the Discussion section, we discussed the possible reasons for the difference in results between NLRP3 and IL-1β. This is described from line 510 to line 517.
Comments 5: The reported significant inverse correlation (Spearman's rho = −0.321, p=0.007) between patient age and the "periodontitis progression risk grade (Grade)" is counter-intuitive. Generally, the severity and stage of periodontitis tend to increase with age due to cumulative disease progression. Clarify precisely whether this correlation refers to the "grade" (risk of progression, A/B/C) or the "stage" (severity, I/II/III/IV). If the finding of an inverse correlation with age for the "Grade" is accurate, provide a detailed biological or clinical explanation for this observation in the "Discussion," considering how the grading system is applied and potential selection biases in the study population (e.g., perhaps older individuals with less aggressive forms were enrolled).
Response 5: Indeed, we report an inverse correlation between age and the grade of periodontitis. In the Discussion section, we have provided an explanation for this result — from line 368 to line 373.
Comments 6: The "Methods" section mentions assessing the "Diagnostic significance of NLRP3 and IL-1β in serum and saliva" using ROC curves. However, the results of this analysis (e.g., Area Under the Curve - AUC, sensitivity, specificity, optimal cutoff values) are not presented in the "Results" section. This analysis is important for evaluating the potential clinical utility of these biomarkers for identifying periodontitis and/or hypertension. Include the results of the ROC curve analysis for NLRP3 and IL-1β (in both serum and saliva), presenting AUC values, confidence intervals, p-values, and potentially suggested cutoff values, in the "Results" section. Discuss the implications of these findings in the "Discussion."
Response 6: The results of the ROC analysis are included — from line 334 to line 336. These results are discussed in the Discussion section — from line 546 to line 557.
Minor Comments:
(Introduction )
Comments 7: The Introduction provides a good background. The transition to discussing NLRP3 and IL-1β could be smoother by briefly summarizing their known roles in periodontitis and hypertension individually before proposing their potential role as mediators linking the two conditions.
Response 7: The role of NLRP3 and IL-1β in periodontitis and hypertension is summarized in the Introduction — from line 99 to line 103.
(M&M)
Comments 8: Specify how participants were recruited from the cardiology clinic (e.g., consecutively enrolled patients meeting inclusion criteria?).
Response 8: It is specified how the participants were recruited for the study — from line 148 to line 151 in the Materials and Methods section.
Comments 9: Clarify if AHA/ACC 2023 and 2018 guidelines were applied interchangeably or if there's a specific rationale for mentioning both years.
Response 9: The diagnosis of hypertension was made in accordance with the 2023 guidelines, which are consistent with those from 2018. This is specified on lines 174–176.
Comments 10: Refine the description of Saliva Collection. Define if the 2 mL target volume was the primary goal, or if the collection strictly adhered to the 10-minute timeframe, regardless of the final volume. "Drool saliva... at least once per minute" could be clarified regarding the intended flow rate (e.g., unstimulated saliva collection usually aims for a specific resting flow rate). Confirm whether the saliva volume was precisely measured for adding the protease inhibitor (1 µL per mL) or if a fixed amount was added to an estimated volume.
Response 10: In the Materials and Methods section, we clarified the description of saliva collection — on lines 212–213. We also specified how the protease inhibitor was added — on lines 216–217.
(Discussion)
Comments 11: Expand the discussion on how the observed elevated NLRP3 levels in the combined disease group specifically align with the known roles of NLRP3 in inflammation and vascular dysfunction related to hypertension.
Response 11: We have discussed how the observed elevated levels of NLRP3 in the comorbidity group correspond to the known roles of NLRP3 in inflammation and vascular dysfunction associated with hypertension – from line 494 to line 498.
Comments 12: Reiterate and emphasize the main limitations of the cross-sectional design clearly in the Discussion.
Response 12: We reiterated and emphasized the main limitations of the cross-sectional study design in the Discussion section – from lines 608 to 611.
Comments 13: While the authors acknowledge other causes of tooth loss (caries, trauma, etc.), briefly discuss if any attempt was made to account for these factors in the data collection or analysis, or acknowledge this as an inherent limitation when using missing teeth as a proxy for periodontitis severity.
Response 13: Accurate information about the cause of tooth loss cannot be obtained. This is an inherent limitation, in principle, when using this indicator to assess the severity of periodontitis.
Comments 14: Conclude the Discussion by suggesting specific directions for future research based on the findings, such as longitudinal studies, intervention trials (e.g., assessing the impact of successful periodontal therapy on blood pressure and these biomarkers), or larger studies capable of more robust multivariable modeling
Response 14: We conclude the discussion by suggesting specific directions for future research – from line 612 to line 615.
Comments 15: The manuscript is generally well-written. However, a final proofread by a native English speaker or a professional editing service would help correct minor grammatical issues and awkward phrasing, enhancing clarity and readability.
Response 15: For the English language correction, we used a professional editing service provided by MDPI.
Round 2
Reviewer 2 Report
Comments and Suggestions for Authors
the authors improve the article